# Desorption of Cadmium from Cocoa Waste Using Organic Acids

**DOI:** 10.3390/foods13244048

**Published:** 2024-12-15

**Authors:** Sebastián Piedrahíta-Pérez, Juliana Rodríguez-Estrada, Margarita Ramírez-Carmona, Leidy Rendón-Castrillón, Carlos Ocampo-López

**Affiliations:** Chemical Engineering Faculty, Centro de Estudios y de Investigación en Biotecnología (CIBIOT), Universidad Pontificia Bolivariana, Medellín 050031, Colombia; sebastian.piedrahita@upb.edu.co (S.P.-P.); juliana.rodrigueze@upb.edu.co (J.R.-E.); margarita.ramirez@upb.edu.co (M.R.-C.); leidy.rendon@upb.edu.co (L.R.-C.)

**Keywords:** cadmium, cocoa waste, cocoa pod husk flour, desorption, leaching, organic acids, citric acid, design of experiments, ORP monitoring

## Abstract

This study evaluated the desorption of cadmium (Cd) from cocoa waste-derived flour using organic acids. Cocoa pods were collected from Antioquia and Casanare, Colombia, to analyze the geographical Cd content and its distribution within the pod tissues. Acid selection was performed using a multi-criteria decision-making (MCDM) matrix, and Cd desorption was assessed through a full factorial 2^3^ experimental design, considering acid concentration, pulp density, and agitation speed. Additionally, the oxidation–reduction potential (ORP) was monitored as an indicator of the electrochemical dynamics of the process. The results indicated that pods from Casanare exhibited higher Cd concentrations (1.63 ± 0.20 ppm) compared with those from Antioquia (0.87 ± 0.22 ppm), with 49.31% of the metal being accumulated in the pod. Parameters of citric acid at 0.5 M, 5 g/L pulp density, and 120 rpm were found to be optimal for the Cd desorption process, achieving over 95% efficiency. Based on ORP monitoring, a heuristic was proposed to determine the contact time during leaching. This work outlines a scalable process for Cd desorption, adding value to cocoa industry waste for potential applications.

## 1. Introduction

Cocoa has established itself as one of the ten most traded products globally [1]. This growth has been driven by the increasing demand for cocoa-derived products and their versatility across various industries, including confectionery, food and beverages, cosmetics, and pharmaceuticals [2,3,4]. In 2022, the global cocoa market exceeded USD 26.6 billion, with a projected annual growth rate of 4.80%, reaching an estimated value of USD 38.8 billion and a global production of 5024 million tons by 2030 [5,6].

In Colombia, cocoa cultivation accounts for 1.5% of the global market, positioning the country as the tenth largest producer worldwide. With approximately 2.8 million hectares available for cultivation, the sector generates annual revenues exceeding USD 23 million and provides employment for 165,000 people [7,8]. Colombia produces approximately 54,785 tons of dry cocoa beans annually, with Santander being the leading producer at 26,315 tons—more than four times the output of the second-largest producer, Antioquia, which produces 5974 tons per year [9].

The cocoa industry generates significant amounts of waste, averaging 20 tons of waste per ton of dry cocoa beans produced, as only 10% of the fruit’s total weight is used commercially. This low utilization rate results in substantial environmental and economic impacts, primarily due to inadequate waste management [10]. In Colombia, approximately 183,412 tons of cocoa pod husks are generated annually and discarded as an agro-industrial byproduct [11].

Despite these challenges, cocoa pod husks contain carbohydrates, fibers, fats, and bioactive compounds such as antioxidants, prompting research into their potential for value-added applications in agriculture, cosmetics, pharmaceuticals, energy, and food industries [12]. For instance, cocoa pod husk flour has been proposed as an additive to traditional flour, enhancing its functional properties [13,14].

However, a major obstacle to utilizing cocoa industry waste is the bioaccumulation of heavy metals, particularly cadmium (Cd). This highly toxic metal occurs naturally in the Earth’s crust due to geological and anthropogenic processes [15]. Its accumulation in plant tissues and subsequent transfer to humans through the food chain poses a significant health threat. Studies have shown that Cd exposure can lead to kidney failure, bone demineralization, and even cancer [16]. Due to its toxic effects, Cd is classified as a hazardous metal alongside lead (Pb), mercury (Hg), and chromium (Cr) [17].

In Colombia, cocoa-growing soils exhibit average Cd concentrations of 1.43 mg/kg across 1837 surveyed soils, a value higher than the global average [18]. This presence of Cd in soils has become one of the most significant challenges to producing safe cocoa in major producing countries [19].

To address this issue, strict regulations have been implemented. Notably, the European Commission Regulation No. 844/2014 establishes tolerable Cd limits in cocoa-derived products, ranging from 0.1 to 0.8 ppm [20]. Similarly, the *Codex Alimentarius* sets a limit of 0.9 ppm of Cd in solid cocoa derivatives, a standard adopted in Colombia through INVIMA regulations [21].

The effective utilization of cocoa waste requires strategies to desorb Cd. Leaching agents for heavy metal removal from solid matrices include inorganic acids, organic acids, chelating agents [22,23], and inorganic substances [24], among others. Organic acids stand out due to their high leaching efficiency under mildly acidic conditions (pH 3–5), achieving over 90% heavy metal removal [25]. Additionally, their biodegradable nature eliminates the need for further treatment of the decontaminated solids, substantially reducing wastewater generation and the environmental impact of desorption processes.

Recently, organic acids have been employed for the leaching of heavy metals from used batteries [25,26], soils [27,28,29], ores [30], catalysts [31], and sludges [32]. However, limited information is available regarding their application to the leaching of food matrices.

Given the above, this study focuses on the desorption of Cd from cocoa agro-industrial waste using organic acids; a tissue analysis of Cd distribution was included, desorption was evaluated through a factorial design of experiments 2^3^, and ORP measurement was proposed as an alternative to monitor the process. This strategy aims to valorize byproducts for high-value applications, contributing to the reduction of environmental impact and promoting a circular economy in the cocoa production chain.

## 2. Materials and Methods

### 2.1. Study Areas and Sample Collection

Cocoa pod waste was collected from two departments in Colombia: Antioquia and Casanare. Antioquia, located in the northwestern region of the country, features a diverse geography ranging from mountainous areas to inter-Andean valleys, which influence cocoa cultivation conditions [33]. Sampling in Antioquia was conducted in several municipalities, including Segovia, Maceo, Bello, Santa Fe de Antioquia, Támesis, Yalí, Mutatá, and Sabanalarga. In contrast, Casanare, situated in the eastern region and characterized by extensive plains and fertile soils [34], provided samples from the municipality of Villanueva. These included various cocoa varieties, such as Luker 40, ICS 1, TCS 01, TSH 565, San Vicente 41, Saravena 13, and FEAR 5.

### 2.2. Cocoa Pod Husk Flour Production

Cocoa pods were peeled, washed, and cut into sections approximately 2 to 3 cm thick. The samples were then dried in a forced convection oven (WTC Binder, 12880, Tuttlingen, Germany) at 72 °C for 48 h [35]. Once cooled, the husks were milled using a blade mill (Waring, 51BL31, Torrington, CT, USA) to obtain particles smaller than 500 µm (Tyler 32 mesh). Additionally, three cocoa pods were randomly selected for a detailed separation of the husk’s components to evaluate the distribution of Cd in each fraction. Figure 1 illustrates the cocoa pod husk flour production process.

### 2.3. Cadmium Quantification by Atomic Absorption

For cadmium (Cd) quantification, four 1 g samples of dry material were subjected to acid digestion using 10 mL of 68% nitric acid and 5 mL of 30% hydrogen peroxide for 60 min, following the EPA Method 3050B [36]. The digested samples were filtered using an 8 µm filter (Cytiva, Whatman 40, Hangzhou, China) and diluted to 25 mL with deionized water.

Cd concentrations were determined by flame atomic absorption spectroscopy (FAAS) using a PerkinElmer AAnalyst 400 spectrometer (PerkinElmer, B3150080, Waltham, MA, USA) with an air–acetylene flame at a ratio of 10/2.5 L·min^−1^, respectively. Absorbance readings were obtained using a Cd hollow cathode lamp (Buck Scientific, 5008, East Norwalk, CT, USA) at a wavelength of 228.8 nm and a current of 4 mA [37].

Absorbance values were correlated with Cd concentration through a least squares linear regression (R^2^ = 0.998) performed beforehand.

### 2.4. Selection of Organic Acid

The selection of organic acids was based on an initial list comprising acetic acid, formic acid, citric acid, oxalic acid, fumaric acid, succinic acid, malic acid, tartaric acid, ascorbic acid, and lactic acid. This selection process employed a multiparametric priority matrix considering the following criteria: commercial cost of the acid (C1), acid dissociation constant (pKa) (C2), compatibility with food matrices (C3), theoretical cadmium leaching capacity in solid matrices (C4), water solubility (C5), and chemical safety (C6). Each criterion was evaluated using a three-level Likert scale (1, 3, 5), as detailed in Table 1.

The weights for each criterion were estimated using the eigenvector technique as described by Rendón-Castrillón et al. [38]. Criterion scores were collected and compared pairwise, applying the Pearson distance correlation, formulated as shown in Equation (1).
(1)$$r\left(x,y\right)=\frac{\sum_{i=1}^{n}\left(x_{i}-\bar{x})(y_{i}-\bar{y}\right)}{\sqrt{\sum_{i=1}^{n}{\left(x_{i}-\bar{x}\right)}^{2}}\sqrt{\sum_{i=1}^{n}{\left(y_{i}-\bar{y}\right)}^{2}}}$$
where x and y represent the scores for the criteria, and $\bar{x}$ and $\bar{y}$ are their respective means.

The correlation data produced a symmetric square matrix R with 6 × 6 dimensions. The eigenvectors of R were calculated using Equation (2).
(2)$$\left(R-\lambda I\right)\cdot W=0$$
where λ represents the eigenvalues of R, I is the 6 × 6 identity matrix, and W is a matrix of eigenvectors of R. The absolute values of the eigenvectors corresponding to the maximum eigenvalue ($\lambda_{max}$) were used as the recommended weights for each criterion.

### 2.5. Cadmium Desorption Kinetics

The optimal desorption time for Cd in cocoa pod husk flour was determined using 1 M citric acid at a pulp density of 5 g/L, with agitation at 200 rpm in a 500 mL glass reactor containing 400 mL of solution. The desorption conditions were selected from previous studies on the metal leaching from soils and ores [25,26,27,31]. Samples of 5 mL were taken at various time intervals over 24 h, filtered, and analyzed to determine the percentage of Cd desorbed using FAAS.

Parallel tests were conducted using acetic, tartaric, malic, and lactic acids, each at a concentration of 1 M for 6 h under identical experimental conditions. An additional test with 0.5 M citric acid was included to evaluate performance at a lower concentration. The acids used here were preselected from the scoring results of the multiparametric priority matrix.

### 2.6. Cadmium Desorption Experiments

The desorption of Cd from cocoa pod husk flour was evaluated using a 2^3^ full factorial experimental design. The independent variables and their levels are shown in Table 2.

A flowchart describing the cadmium desorption experiments is shown in Figure 2.

The contact time was fixed at 6 h at room temperature, and the response variable was the percentage of Cd desorbed from the flour. The 2^3^ factorial design matrix is shown in Table 3.

Each experiment was performed in duplicate in 500 mL glass reactors with an effective volume of 250 mL and mechanical agitation. After the contact time, the Cd concentration in the solution was quantified using FAAS.

### 2.7. Mathematical Modeling of Desorption

Cd desorption was evaluated under optimal conditions identified through the 2^3^ factorial design. During this process, the oxidation–reduction potential (ORP) was monitored using an ORP electrode (SCHOTT Instruments, PT6880, Mainz, Germany), alongside the Cd concentration in the solution.

ORP monitoring data were fitted to a nonlinear predictive regression model. This was achieved using a scalar function regression routine in two dimensions (2D), implemented in Python. The optimization objective was used to minimize the sum of absolute error squares for the model fit.

### 2.8. Statistical Analysis

An ANOVA test was performed to identify significant differences (*p* < 0.05) between treatments using Statgraphics Centurion XIX^®^ (Statgraphics Technologies, Version 19.0, The Plains, VA, USA). Data visualization was carried out using OriginPro 2024b (OriginLab Corporation, Version 10.1.5.132, Northampton, MA, USA). Maps and illustrations were created with Affinity Designer 2 (Serif Europe Ltd., Version 2.5.5, Nottingham, UK) and QGIS Grenoble (QGIS Development Team, Version 3.38.3, Grüt, Switzerland).

## 3. Results and Discussion

### 3.1. Cadmium Content in Cocoa Pods: Geographic Analysis

The geographic analysis of Cd content in cocoa pods collected from two Colombian departments (Antioquia and Casanare) revealed significant variations in the concentration of this heavy metal between the regions. Figure 3 illustrates the geographic distribution of Cd in the cocoa samples.

Cd concentration data showed significant differences between the departments. In Antioquia, Cd levels ranged from 0.460 to 1.227 ppm, with a mean of 0.868 ppm and a standard deviation of 0.205. Conversely, Casanare presented higher Cd concentrations, ranging from 1.402 to 1.928 ppm, with a significantly higher mean of 1.627 ppm and a standard deviation of 0.181. These results suggest that cocoa from Casanare tends to accumulate higher Cd levels than that from Antioquia, potentially due to region-specific geographic and edaphic factors [39].

Within Antioquia, inter-municipal variability was lower compared with Casanare. Cd concentrations in Antioquia showed broader dispersion, with the lowest value being in Segovia (0.460 ppm) and the highest being in Támesis (1.227 ppm). This range indicates that certain areas, such as Támesis and Maceo, may have conditions favoring Cd absorption [19]. Factors such as soil composition and the use of agrochemicals may influence this variability, although mean concentrations remained within permissible levels compared with other regions of Colombia [40].

The results from Antioquia align with Cd levels reported in soils from 60 farms in the department, which range between 0.2 and 1.86 ppm, with an average of 1.14 ppm [19]. Comparing these soil values with the Cd concentrations in cocoa pods suggests that not all available soil Cd is absorbed by cocoa plants. The observed correlation highlights the influence of soil composition and interactions between Cd and other elements, which may limit its bioavailability for uptake by cocoa roots [41]. Similarly, Cd levels in cocoa beans from Maceo range from 0.07 to 1.44 ppm, corresponding to the husk values reported in this study for the same locality [33].

In Casanare, particularly in the Villanueva municipality, Cd levels were notably higher. The varieties Saravena 13 and TSH 565 exhibited the highest concentrations (1.928 ppm and 1.840 ppm, respectively), exceeding current food safety regulations. The lower variability in Casanare, reflected by a standard deviation of 0.181, suggests greater uniformity in Cd concentrations, potentially driven by consistent soil and environmental factors in the region [42]. This uniformity may indicate steady bioaccumulation influenced by soil mineral composition or agricultural practices. Moreover, the high Cd levels in Casanare could be associated with specific geographic conditions, as neighboring departments like Cundinamarca, Boyacá, Arauca, and Santander also exhibit elevated Cd levels in their soils [18,43,44].

From a geological perspective, soil formations play a critical role in Cd bioavailability. In Colombia, the distribution of this heavy metal is influenced by geological terrains, and the differences observed in this study align with geochemical trends associated with the lithology of the analyzed locations [45]. In Casanare, sedimentary formations, including conglomerates and sandstones, along with predominantly alluvial soils, favor the presence of minerals that release Cd in more soluble forms, increasing its bioavailability for plant uptake [46,47]. In contrast, Antioquia’s geology is more varied and complex, featuring formations such as the Antioquia Batholith and sedimentary rocks like sandstones, litharenites, and micrites, which contain quartz and minerals that tend to immobilize Cd [46,47]. Additionally, the higher organic matter content in Antioquia’s soils may further restrict Cd bioavailability, limiting its absorption by cocoa plants [45].

### 3.2. Cadmium Distribution in Cocoa Pod Components

The cocoa pod was analyzed to determine the tissue-specific distribution of Cd, focusing on the epicarp (outer layer), mesocarp (middle layer), scleral layer (hardened part between the mesocarp and endocarp), endocarp (innermost layer covering the beans), and the interior containing the cocoa beans [48]. This segmentation provided insights into how Cd mobilizes and accumulates within different tissues, elucidating its bioaccumulation dynamics and implications for cocoa waste utilization after bean extraction. Figure 4 presents the pod’s anatomical components, specifying the dry weight fraction, Cd concentration, and the percentage contribution of Cd to the entire pod, based on the average values from three analyzed pods.

The epicarp exhibited a Cd concentration of 1.23 ppm, accounting for 6.35% of the total Cd content in the pod based on dry weight. Despite being the pod’s outermost barrier, the epicarp contributes minimally to overall Cd accumulation. This limited retention can be attributed to its direct exposure to external contaminants and reduced transport capacity toward the interior [49].

The mesocarp showed a Cd concentration of 0.96 ppm, contributing 15.78% of the total Cd in the pod. This intermediate accumulation indicates that the mesocarp serves as a transitional storage zone for Cd, with lower levels compared with internal tissues.

The scleral layer had a Cd concentration of 1.31 ppm, representing 3.21% of the total Cd in the pod. Despite a similar concentration to the mesocarp and endocarp, its smaller relative mass limits its overall Cd contribution. This structural layer’s accumulation is likely influenced by its limited role in vascular transport and Cd mobility, which depends on factors such as solubility and the presence of compounds facilitating Cd transport [50,51].

The endocarp, the innermost layer surrounding the cocoa beans, exhibited a Cd concentration of 1.32 ppm, accounting for 29.97% of the pod’s total Cd. This high proportion highlights the endocarp as a critical accumulation zone, reflecting Cd translocation and deposition before reaching the beans [52]. The endocarp’s role as a protective barrier and its involvement in vascular transport contribute to its significant Cd concentration [53].

The interior, containing the cocoa beans, had the highest Cd concentration at 1.55 ppm, representing 50.69% of the total Cd in the pod. This suggests that Cd permeates the outer layers and accumulates in the beans, which are the consumable product [54,55]. The placenta, a tissue connecting the beans to the plant, may facilitate this accumulation by transporting nutrients and other compounds, including Cd, through the vascular system into the developing beans [51].

The high Cd content in the beans raises concerns due to their primary use in cocoa production, emphasizing the need to monitor and reduce Cd exposure in plantations to ensure product safety [56,57]. Additionally, the Cd content in the pod’s husk, accounting for 49.31% of the total Cd in the fruit, underscores the importance of managing Cd levels in the husk. Reducing Cd in the husk is critical not only for food safety but also for enabling the safe use of cocoa waste in various industries [58,59].

### 3.3. Multiparametric Priority Matrix for Organic Acid Selection

To evaluate the Cd desorption process, a multiparametric priority matrix was developed, enabling the comparison of various organic acids based on the criteria defined in Section 2.4. Using the scores for criteria C1 to C6, a Pearson correlation coefficient matrix, R, of a 6 × 6 dimension was constructed, as presented in Equation (3).
(3)$$R_{6\times 6}=\left[\begin{array}{cccccc} 1.000 & 0.241 & -0.472 & 0.709 & -0.040 & -0.373\\ 0.241 & 1.000 & -0.452 & 0.218 & -0.167 & -0.310\\ -0.472 & -0.452 & 1.000 & -0.263 & 0.536 & 0.748\\ 0.709 & 0.218 & -0.263 & 1.000 & 0.024 & -0.406\\ -0.040 & -0.167 & 0.536 & 0.024 & 1.000 & 0.069\\ -0.373 & -0.310 & 0.748 & -0.406 & 0.069 & 1.000 \end{array}\right]$$

Eigenvalues and eigenvectors were calculated using matrix estimation algorithms. The eigenvalues of matrix R are shown in Equation (4), and the eigenvectors corresponding to the maximum eigenvalue $\lambda_{max}$ = 2.788 are presented in Equation (5).
(4)$$\lambda_{R}=\{\begin{array}{cccccc} 2.788 & 1.325 & 0.054 & 0.327 & 0.792 & 0.715 \end{array}\}$$
(5)$$W={\left[\begin{array}{cccccc} -0.442 & -0.340 & 0.519 & -0.397 & 0.210 & 0.467 \end{array}\right]}^{T}$$

By normalizing the eigenvectors, the weights of criteria C1 to C6 were determined, as shown in Equation (6).
(6)$$W\prime ={\left[\begin{array}{cccccc} 0.186 & 0.143 & 0.219 & 0.167 & 0.088 & 0.197 \end{array}\right]}^{T}$$

The multiparametric priority matrix for the selection of the organic acid is presented in Table 4, which shows the scores for each criterion according to literature reports, the score obtained, and the position compared with the other acids evaluated.

The multiparametric matrix identified citric acid as the most suitable option for Cd desorption from cocoa waste. Among the ten acids evaluated, citric acid achieved the highest overall score of 4.71, excelling in five of the six criteria assessed. Citric acid has been extensively studied as a leaching agent in various solid matrices, demonstrating excellent results in the removal of metals such as Cd, Pb, Zn, Ni, Cr, Co, Al, Li, Mn [25,60,61,62,63,64,65,66,67], and even rare earth elements like Sc, Y, La, Ce, Pr, Nd, Sm, Gd, and Yb [30,71]. Its chemical properties, affordability, safety features, and widespread use in the food industry [67] make it an optimal choice for processes where preserving the integrity of the food matrix is critical, as in the case of flour derived from cocoa residues. Furthermore, citric acid’s high solubility facilitates its application in aqueous media [60]; additionally, its favorable safety profile minimizes handling risks and enhances process scalability [72].

The global analysis of the matrix ranked the top five acids as follows: citric acid (4.71), acetic acid (4.32), tartaric acid (4.29), malic acid (4.01), and lactic acid (3.63). These acids represent a balance between cost, desorption efficiency, and safety. Acetic acid, despite having a lower dissociation constant than other acids [25], is highly compatible with food matrices [66] and is notable for its availability and low cost [62]. Tartaric acid emerges as an effective alternative due to its high water solubility and leaching capability [61], while malic and lactic acids, with slightly lower scores, offer good compatibility with food matrices and excellent chemical safety [67], making them viable options depending on specific process requirements.

Notably, the criteria C3 (compatibility with food matrices) and C6 (chemical safety) received the highest weights in the matrix, at 21.9% and 19.7%, respectively. These criteria underscore the importance of selecting an acid that does not compromise food product quality while maintaining an appropriate safety profile for industrial use [78]. The high weight of C3 highlights the need to ensure that the acid employed does not interfere with the organoleptic properties or quality of cocoa husk flour. Meanwhile, the emphasis on C6 reflects the relevance of using acids with low toxicity and minimal environmental risk, which is particularly important in the food industry and in industrial waste management [79].

The acids with the lowest scores in the matrix were formic acid (2.78), oxalic acid (2.99), and succinic acid (3.28). These acids exhibited significant limitations across several key criteria. This analysis suggests that, although certain acids may demonstrate technical efficacy, their limitations in terms of safety, commercial cost, and compatibility with food matrices are critical factors leading to their exclusion from such processes.

In conclusion, citric acid stands out as the optimal choice for Cd desorption from cocoa residues, followed by other acids with complementary characteristics. The application of this priority matrix ensures that the selection of the organic acid is the most appropriate in terms of cost-effectiveness, safety, and sustainability for heavy metal desorption processes.

### 3.4. Desorption Kinetics

The optimal desorption time for Cd removal from cocoa residue flour using the identified organic acids was determined, as shown in Figure 5, which illustrates Cd desorption with 1 M citric acid over a 24 h period.

An initial rapid increase in the percentage of Cd removal was observed within the first 180 min (3 h), reaching nearly 60% desorption. Thereafter, the desorption rate slowed and stabilized at approximately 95% by 360 min (6 h), suggesting that most of the Cd had been removed by this time. The slight increase between 360 and 1440 min indicates that equilibrium in leaching capacity had been reached with citric acid. This typical kinetic behavior—characterized by an initial rapid phase followed by a plateau—is common in desorption processes, where easily accessible Cd sites are removed quickly, while the remaining fraction is more tightly bound to the solid matrix [80].

The efficacy of citric acid as a leaching agent can be attributed to its strong chelating properties, forming stable complexes with heavy metals like Cd. Its structure, which contains three carboxyl groups and one hydroxyl group, enables the formation of bonds with metal ions, facilitating their mobilization and desorption from solid matrices such as cocoa husks [81].

The key outcome of this experiment is the identification of 6 h as the optimal desorption time, as no significant increase in metal removal is observed beyond this point. This time is crucial for the experimental design of subsequent trials, maximizing process efficiency by reducing operational time and optimizing costs in industrial applications. Additionally, the high efficacy of 1 M citric acid, achieving nearly 100% Cd desorption, validates its suitability as a leaching agent for this type of matrix [82,83].

Figure 6 presents Cd desorption kinetics for acetic, tartaric, malic, and lactic acids, as well as citric acid at 0.5 M, over a 6 h period. The results demonstrate the superior performance of citric acid, both at 1 M and 0.5 M concentrations, compared with the other acids. Citric acid at 1 M achieved almost 100% desorption, followed by tartaric acid (1 M) and lactic acid, which reached approximately 85% and 73%, respectively, after 360 min. Acetic and malic acids exhibited intermediate behavior, achieving about 50% desorption, with acetic acid being the least effective in this context. Notably, even at a lower concentration (0.5 M), citric acid remained highly effective, outperforming all other acids except tartaric acid, which achieved a similar removal percentage.

The analysis of these desorption kinetics highlights the influence of both acid concentration and intrinsic properties on desorption capacity [22]. Citric acid demonstrated superior efficiency across both high and low concentrations, consistent with its theoretical ability to form stable complexes with heavy metals like Cd [60]. Tartaric acid, which was not prominent in the multiparametric matrix, exhibited unexpectedly strong leaching capabilities, positioning it as a competitive alternative. While acetic, lactic, and malic acids were effective, they required longer times to achieve maximum desorption capacity, which could limit their industrial application in processes demanding high throughput and short operational times [22].

A comparative analysis of Cd desorption data from this study with results for other metals reported in the literature, such as Co [60], Cu, and Cr [61], where organic acids were employed, reveals variability in removal efficiency and rate depending on the metal type, acid, and concentration. For instance, malic and acetic acids showed relatively short desorption times for Cd and Co, but greater efficiency for Co removal. Similarly, acetic acid achieved desorption percentages of approximately 60% for Cu and 20% for Cr within 3 h at the same concentrations used in this study [61]. Citric acid, on the other hand, demonstrated significant metal-specific dependencies, being particularly effective at Cd removal within substantially shorter times [60]. These findings underscore the importance of considering the metal’s nature, the solid matrix’s characteristics, and experimental conditions when selecting an organic acid for desorption processes.

### 3.5. Evaluation of Cadmium Desorption

Cd removal from cocoa residues was evaluated using a full factorial 2^3^ design to identify the influence of three variables: citric acid concentration, pulp density, and agitation speed. Time and temperature were fixed, with the latter being set at ambient conditions to minimize operational costs and energy consumption [84]. The desorption time was established at 6 h based on the kinetics described in Section 3.4.

As for the independent variables of the process, citric acid was used at the concentrations evaluated in the previous section (0.5 and 1 M). The mass concentration of the cocoa sample or pulp density was evaluated within a range between 5 and 10 g/L as these are moderate concentrations that allow the leaching agent to effectively penetrate the solid matrix of the cocoa meal without saturating it, maximizing its interaction with the heavy metal [85]. Finally, the agitation speed is evaluated between 80 and 120 rpm, which ensures a uniform movement of the particles without affecting the measurement or causing shear effects.

Table 5 shows the experimental results of the percentage of Cd desorption for each configuration of the factorial design 2^3^, with its respective replication and the random order being established. The factors analyzed were citric acid concentration (A), pulp density (B), and agitation speed (C), represented in their upper (+1) and lower (−1) levels.

Statistical analyses, including estimated effects, Student’s *t*-test, analysis of variance (ANOVA), F-test, and optimal factor values, were performed using specialized software (Statgraphics Centurion XIX^®^, The Plains, VA, USA).

#### 3.5.1. Estimated Effects

Table 6 displays the estimated effects of each factor and their linear interactions. Errors were calculated based on the total error with eight degrees of freedom (block number omitted).

The signs of the estimated effects indicate the levels at which higher Cd removal occurs. For acid concentration and pulp density, a change from the higher level (+1) to the lower level (−1) results in increases of 9.01% and 37.74% in desorption efficiency, respectively. For agitation speed, a change from the lower level (−1) to the higher level (+1) results in a 9.06% increase in efficiency. Interactions are less straightforward to interpret as they involve contrasts rather than isolated effects.

#### 3.5.2. Student’s *t*-Test

To assess the statistical significance of the estimated effects, a Student’s *t*-test was performed. At a 95% confidence level (α = 0.05) with eight degrees of freedom, the critical *t*-value for a two-tailed test is 2.3. Effects with a standardized value (effect/error ratio) exceeding this threshold were deemed statistically significant.

Figure 7 shows a Pareto chart of standardized effects, with factors and interactions being ranked by their absolute values. The vertical line represents the critical *t*-value. Factors exceeding this line are statistically significant. For this analysis, pulp density and acid concentration were significant at their lower levels, while agitation speed was significant at its higher level. The interaction between acid concentration and agitation speed (AC) was significant, though the analysis of its specific levels is less direct.

In Figure 7, the blue bars represent the factors, where the negative sign corresponds to their lower evaluated levels. Similarly, the orange bar indicates that the factors are statistically significant when evaluated at their higher levels.

#### 3.5.3. ANOVA: F-Test

Table 7 shows the information obtained from the ANOVA test for the design of experiments.

The analysis of variance (ANOVA)–F-test partitions the variability of the desorption percentage, thereby testing the statistical significance of each effect by comparing the test statistic (F_0_) with the F-value. The F_0_-value is the ratio of the respective mean square to the mean square of the experimental error. At a 95% confidence level (α = 0.05), with one degree of freedom for the effects and eight degrees of freedom for the error, the F-value for a one-tailed right distribution is 5.32. If the test statistic F_0_ is greater than the F-value, the mean square of the effect is significantly larger than that of the error, leading to the conclusion that the effect is statistically significant.

For the obtained data, all of the test statistics for the individual factors are greater than the F-value, indicating statistical significance. However, the interactions between acid concentration and pulp density (AB) and between pulp density and stirring speed (BC) are not statistically significant as their test statistics are smaller than the critical F-value. In contrast, the interaction between acid concentration and stirring speed (AC) is statistically significant. These results are consistent with those obtained from the *t*-test.

Additionally, linear regression yields the R^2^ statistic, indicating that the adjusted model explains 98.69% of the variability in Cd desorption. It is worth mentioning that the adjusted R^2^, which is more appropriate for comparing models with different numbers of independent variables, is 97.54%.

#### 3.5.4. Optimum Values of the Factors

According to the full factorial design of experiments 2^3^, the relevant values of the factors studied, understood as the values at which the highest Cd desorption occurs, are shown in Table 8.

From the model derived from the experimental design, it is expected that a desorption efficiency of 94.73% for Cd from cocoa pod husk flour can be achieved under these optimal conditions.

The results show that better Cd desorption is achieved when using a low concentration of citric acid, as this condition maximizes desorption efficiency. This can be explained by the fact that, at higher concentrations, citric acid tends to form stable complexes with heavy metals, including Cd, which interferes with analytical measurements, as highlighted by recent studies on the chelating capacity of this acid in aqueous media [86]. Furthermore, while kinetic experiments demonstrated that a citric acid concentration of 1 M produced the highest desorption results, reducing the concentration to 0.5 M offers significant advantages at an industrial level. This reduction translates into lower operational costs, as only half the amount of organic acid is needed to prepare the leaching solution. This not only decreases raw material consumption but also reduces costs associated with storage, handling, and waste disposal.

Although citric acid is an accessible and effective reagent for leaching metals like Cd, higher concentrations do not necessarily improve the treated material. Moderate concentrations of organic acids like citric acid are sufficient to mobilize heavy metals without damaging the original matrix of the treated material—in this case, cocoa—thereby promoting controlled extraction [27].

Regarding pulp density, lower solid concentrations of cocoa improve desorption results. This phenomenon has been documented in studies on metal leaching, where higher solid content can inhibit the interaction between the acid and Cd adsorption sites. The saturation of the active sites of the matrix with the leaching agent, as well as reduced mass transfer efficiency, are the main factors explaining the lower removal efficiency under high solid concentration conditions [85]. Additionally, the limited dispersion of Cd in high pulp density systems may be influenced by increased viscosity and diffusion limitations within the heterogeneous system.

Finally, stirring speed must be high enough to keep solids in suspension and prevent settling, thereby enhancing reaction kinetics. At appropriate stirring speeds, diffusional resistances surrounding the particles are reduced, promoting faster mass transfer and consequently more efficient Cd removal while also reducing operational costs [84].

### 3.6. Monitoring via ORP

The experiment was conducted under the optimal conditions previously established through the experimental design. During the trial, ORP monitoring was performed continuously for 6 h, with an additional single ORP measurement at 24 h to determine the stabilization point of the potential. Simultaneously, Cd desorption was quantified using FAAS. This monitoring, shown in Figure 8, aimed to understand the electrochemical dynamics associated with Cd release from the solid matrix and to evaluate potential correlations between ORP changes and Cd desorption.

At the beginning of the experiment, the ORP starts at 400 mV and shows a slight decrease during the first few minutes, reaching its minimum value after approximately 30 min. This initial decline may be associated with changes in the chemical species present in the solution during the initial contact between the acid and the solid matrix, potentially due to the reduction of metallic species linked to Cd release [87]. Subsequently, the ORP steadily increases, stabilizing around 410 mV after approximately 6 h (360 min). This stabilization coincides with the point where Cd desorption reaches its maximum value, approximately 95%. This gradual rise in ORP suggests increased oxidation in the solution, likely associated with the progressive desorption of Cd from the solid matrix into the aqueous solution.

In contrast, Cd desorption exhibits a more rapid increase during the first 100 min, achieving nearly 60% removal during this period. Afterward, the removal kinetics slow, with a more gradual and continuous increase until reaching 95.2%. Notably, the Cd desorption results validate the model defined through the experimental design, with the removal efficiency exceeding the predicted value under optimal conditions. The divergence between the ORP and Cd desorption curves indicates that Cd desorption is more efficient in the early stages of the process, whereas the increase in ORP reflects chemical stabilization in the system as most of the metal has already been extracted.

ORP monitoring demonstrates significant potential for optimizing metal desorption processes by providing real-time information. This approach could substantially reduce operational costs by minimizing the need for frequent and costly quantification analyses such as FAAS [88]. ORP monitoring as an indicator has been explored in water treatment processes [89], enzymatic processes [90], cell cultures [91], electrochemical reactions [88,92], and metal removal processes for elements such as Cr [93] and Fe [94]. However, no direct relationship with Cd desorption has been previously verified, making this study a novel approach for employing ORP as an indicator in the Cd leaching process from solid matrices.

Based on the results obtained, ORP was identified as a relevant variable for determining the optimal Cd desorption point in the evaluated samples. While previous kinetic studies established a fixed time for achieving maximum desorption, continuous ORP monitoring offers an additional advantage by enabling real-time identification of the point of maximum desorption efficiency. This allows for dynamic adjustment of operational times, optimizing the desorption process and avoiding unnecessary reaction times that could increase operational costs.

Thus, ORP as a function of time was fitted to a predictive nonlinear regression model using a two-dimensional (2D) scalar function regression routine. This routine defined that the data could be represented by a double asymptotic exponential B with offset function, as presented in Equation (7). The model parameters are summarized in Table 9.
(7)$$ORP(t)=a\left(1-e^{bt}\right)+c\left(1-e^{dt}\right)+Offset$$

From the model, the parameters and their influence on the fitting curve are analyzed. The parameter a, associated with the first exponential term of the model, is related to the decrease in ORP at the beginning of the desorption process. The rapid decrease in ORP values during metal desorption processes is consistent with experiments conducted by Yu et al. [93]. The value of b controls the rate at which the first exponential term approaches its final value. Because b is negative and small in absolute value, it indicates that the initial ORP drop is slow, meaning the system takes time to stabilize during this first phase, lasting approximately 30 min.

The parameter c, which has a positive sign, is associated with the second phase of the model. It represents the growth or increase in ORP over time after the initial decrease described by parameter a. A positive and relatively high value of *c* indicates a significant increase in ORP after the initial phase, with c serving as an approximation of the distance between the global minimum and the asymptote of the model. This increase may be linked to the release of Cd as desorption progresses, with ORP values reflecting the evolution of the chemical process, as confirmed by Cd desorption values obtained through FAAS.

The parameter d, accompanying the second exponential term, describes the rate at which the ORP stabilizes after the increase driven by c. With a very small and negative value, d indicates that the second increase is slow and gradual, suggesting a prolonged stabilization phase. This may imply that, during the final stages of the desorption process, the system gradually approaches a steady state, consistent with behaviors observed in desorption systems [95].

Lastly, the Offset term represents the reference value or starting point of the ORP when time equals zero. In this case, the value of 398.759 mV serves as the starting point of the redox potential at the beginning of the experiment, close to the actual measured value of 400 mV. The model parameters also allow for the determination of the ORP value at which the readings stabilize over time, approximating the value of a+c+Offset.

The comparative graph between the experimental data and the model fitting, also shown in Figure 8, illustrates that the model closely follows the observed trend, capturing both the initial ORP drop and its subsequent increase and stabilization.

To interpret the behavior of ORP as a function of time and the Cd desorption process, the proposed model was used to define the desorption time based on the intersection between the model’s asymptote and a tangent line at a specific point on the ORP–time curve. This approach was inspired by the Ziegler and Nichols tuning method [96,97], as shown in Figure 9.

In Figure 9, the asymptote represents the limit that ORP approaches as time tends toward infinity, indicating the final steady state of the desorption process, as expressed in Equation (8).
(8)$$\underset{t\to \infty }{\lim}ORP\left(t\right)=a+c+Offset$$

The tangent line was calculated at the point where the ORP returns to its initial measured value at time zero, following an initial decrease and sustained increase. This point in time is referred to as θ. The return of ORP to the initial value represents the system’s equilibrium, where the desorption process restores the initial potential after the disturbance caused by the process onset. The slope of the tangent line at this point was determined using the derivative of the ORP model with respect to time, as shown in Equation (9).
(9)$${\left.\frac{d\;ORP\left(t\right)}{dt}\;\right|}_{t=\theta }=-{abe}^{b\theta }-{cde}^{d\theta }$$

The intersection between this tangent line and the asymptote defines the time parameter (τ), which indicates the optimal duration of the desorption process to achieve maximum Cd removal from the sample. The values of parameters θ, τ, and their ratio are presented in Table 10.

The heuristic proposed for the optimal desorption time is given in Equation (10).
(10)τ=λθ→τ=2.31θ

The heuristic approximation obtained through the analysis of τ provides a simple and effective methodology for identifying the optimal desorption time without relying on sophisticated analytical techniques such as FAAS. This ORP-based approach offers a practical tool for the rapid and non-invasive estimation of desorption efficiency, which is beneficial for in situ monitoring applications and preliminary optimization studies for heavy metal extraction processes [98].

## 4. Conclusions

This study presents a novel, scalable approach for cadmium (Cd) desorption from cocoa residues using citric acid, achieving over 95% removal of this heavy metal. This enables the repurposing of byproducts from the cocoa production chain for other potential applications.

The results reveal that Cd concentration in cocoa pods from Casanare (1.63 ± 0.20 ppm) is significantly higher than in those from Antioquia (0.87 ± 0.22 ppm), suggesting geographical and edaphological influences on metal accumulation. Furthermore, tissue analysis indicated that 49.31% of Cd accumulates in the pod, limiting the use of these residues in other industries.

Citric acid was identified as the most effective organic acid for Cd removal from cocoa residues based on the MCDM matrix results and kinetic desorption studies using different acids and concentrations.

A full factorial experimental design evaluated the effects of acid concentration, pulp density, and stirring speed on Cd desorption. A first-order effects analysis demonstrated statistical significance for all variables, with pulp density being the most relevant for Cd desorption. Optimal conditions were determined to be citric acid at 0.5 M, a pulp density of 5 g/L, and a stirring speed of 120 rpm, achieving 95.2% Cd desorption.

Monitoring the oxidation–reduction potential (ORP) proved to be a valuable tool for estimating the duration of the desorption process, providing a real-time in situ control method that can be implemented in future industrial applications for heavy metal desorption.

## Figures and Tables

**Figure 1 foods-13-04048-f001:**
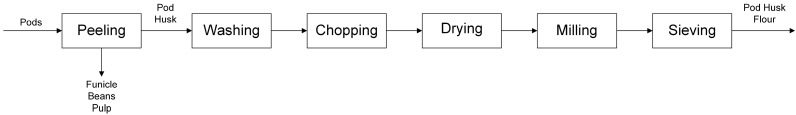
Flowchart of the flour production process.

**Figure 2 foods-13-04048-f002:**
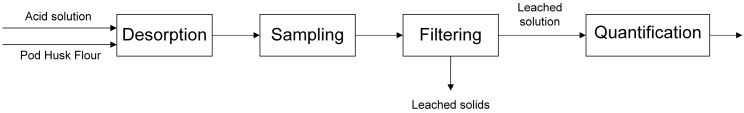
Flowchart for the cadmium desorption experiments.

**Figure 3 foods-13-04048-f003:**
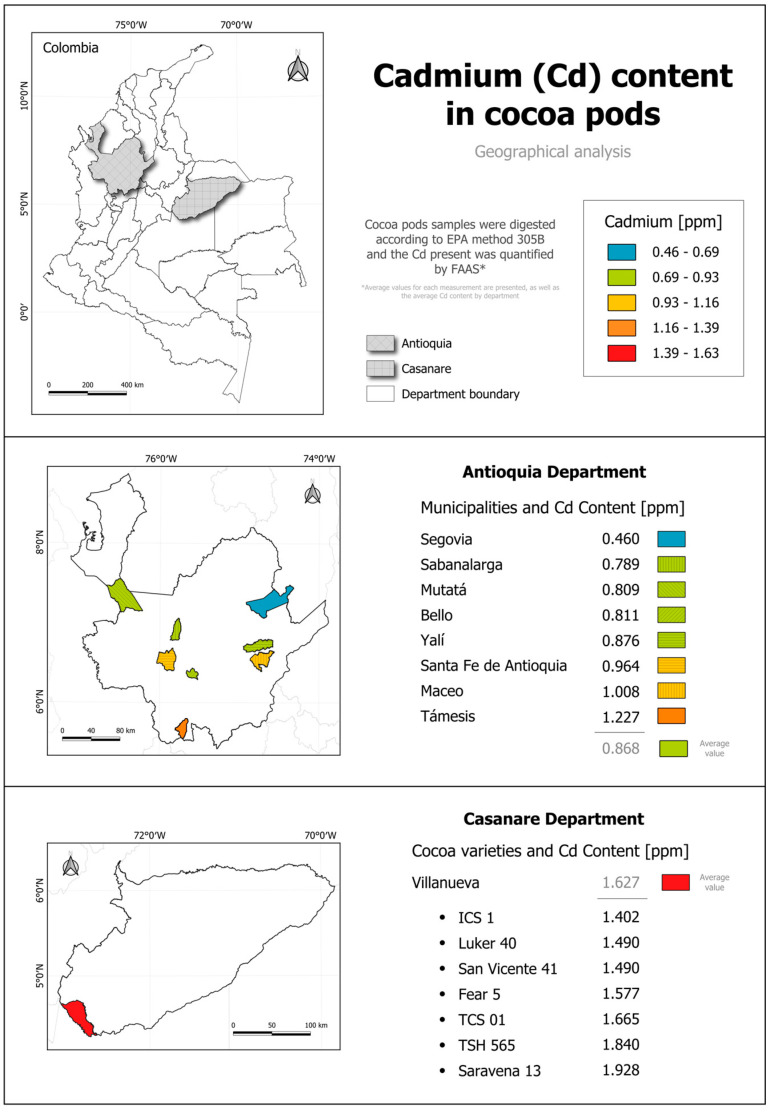
Cadmium (Cd) content in cocoa pods from Antioquia and Casanare, Colombia. *Average values for each measurement are presented, as well as the average Cd content by department.

**Figure 4 foods-13-04048-f004:**
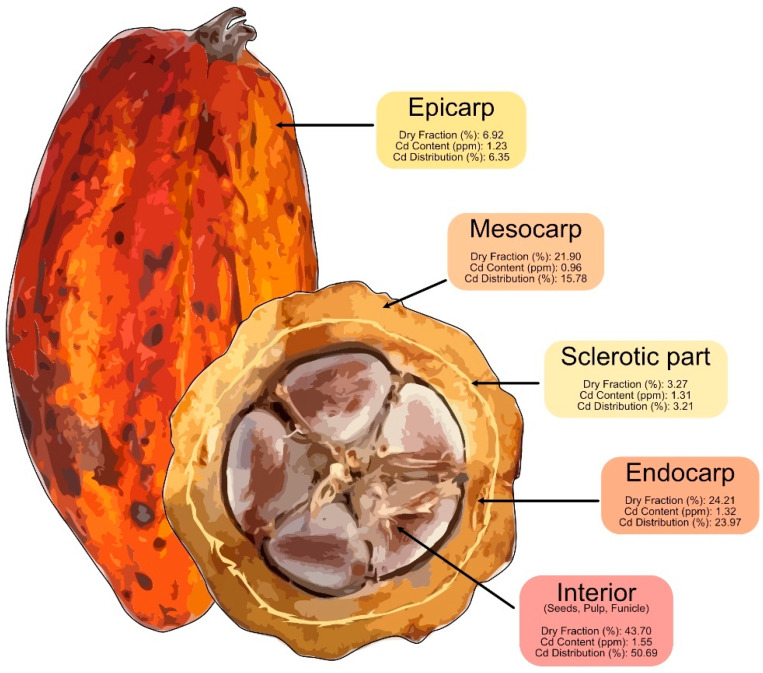
Tissue-specific analysis of cadmium (Cd) distribution in the cocoa pod.

**Figure 5 foods-13-04048-f005:**
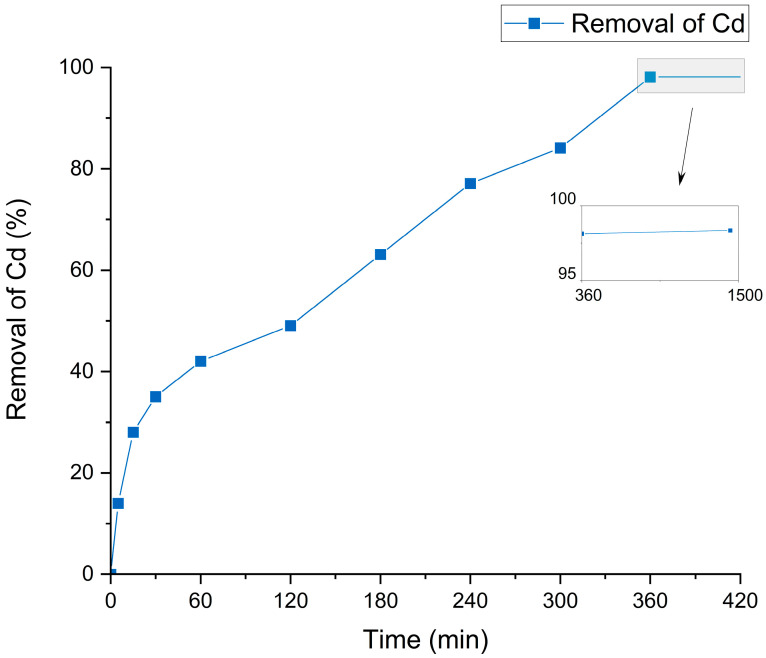
Cd desorption kinetics from cocoa residue flour using 1 M citric acid over 24 h. The main graph covers the first 6 h, while the secondary graph covers the period from 6 to 24 h.

**Figure 6 foods-13-04048-f006:**
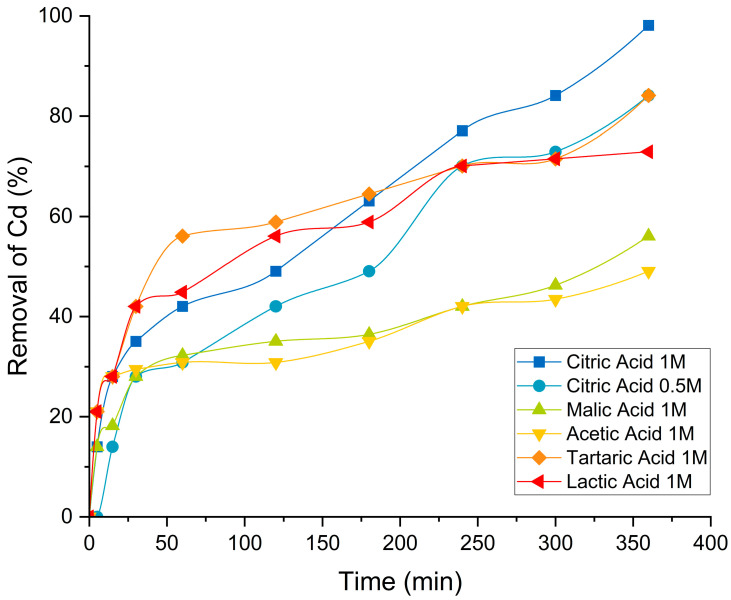
Cd desorption kinetics from cocoa residue flour using organic acids.

**Figure 7 foods-13-04048-f007:**
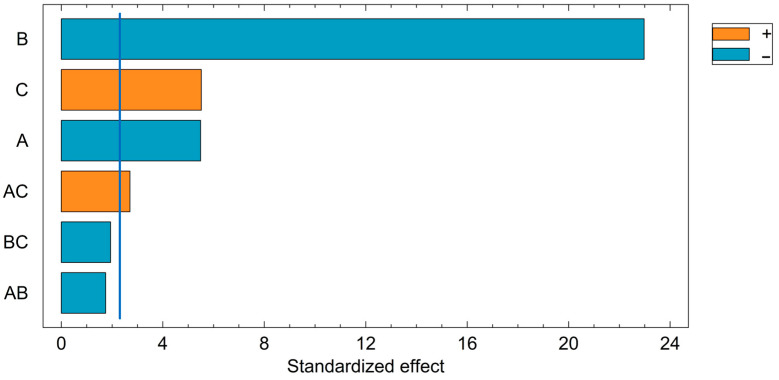
Pareto chart of standardized effects.

**Figure 8 foods-13-04048-f008:**
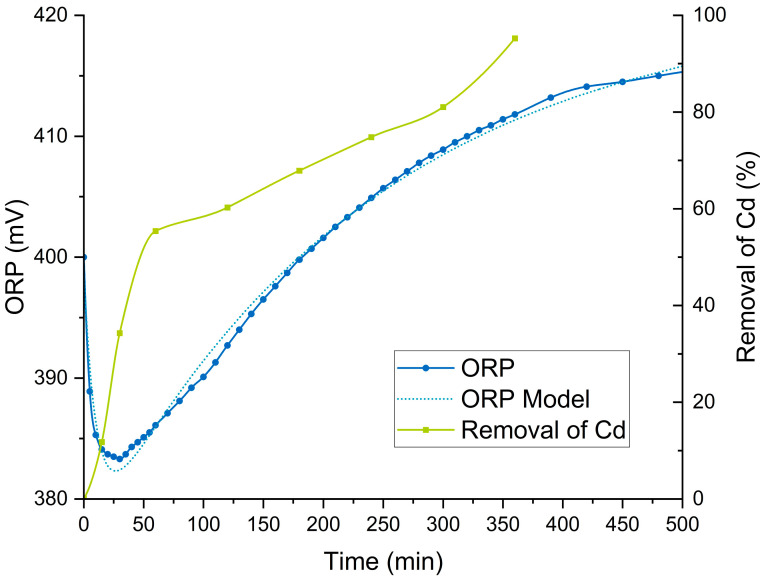
ORP and Cd monitoring in a desorption trial of cocoa waste. The left axis shows ORP in millivolts (mV), while the right axis represents the percentage of Cd removal.

**Figure 9 foods-13-04048-f009:**
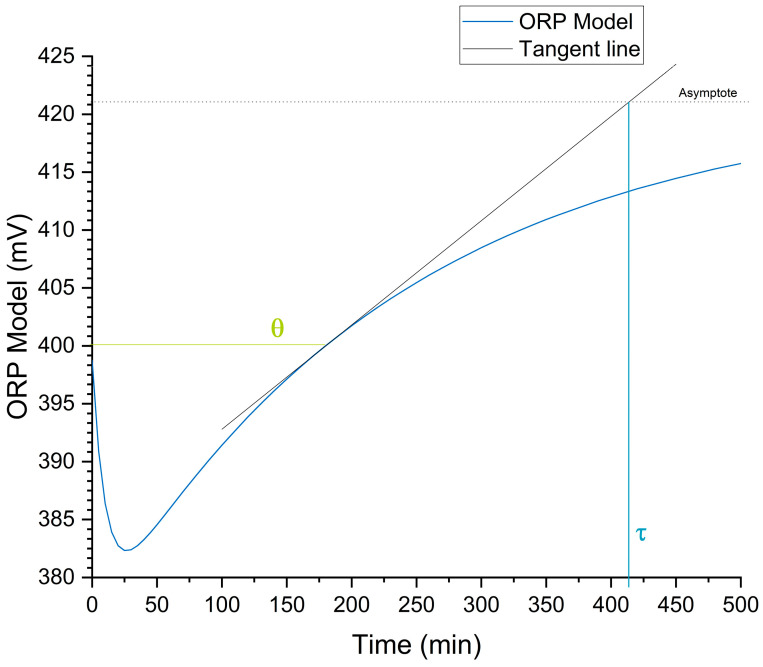
Deduction of heuristic parameters in the Cd desorption process based on ORP.

**Table 1 foods-13-04048-t001:** Criteria for organic acid selection using a Likert scale.

Criterion	1	3	5
1. Commercial cost of the acid	Expensive acid compared with other market options. >20 USD/kg	Moderately priced relative to available alternatives. Between 10 and 20 USD/kg	Economical acid, offering a clear economic advantage. <10 USD/kg
2. Acid dissociation constant (pKa)	Unsuitable pKa for the application. >5.0	Moderately suitable pKa. Between 3.0 and 5.0	Optimal pKa for the application. <3.0
3. Compatibility with food matrices	Low compatibility with food matrices.	Moderate compatibility with food matrices.	High compatibility with food matrices.
4. Theoretical cadmium leaching capacity	Low leaching capacity, low affinity for Cd complex formation.	Moderate leaching capacity, moderate affinity for Cd complex formation.	High leaching capacity, high affinity for Cd complex formation.
5. Water solubility	Low water solubility. <100 g/L	Moderate water solubility. Between 100 and 350 g/L	High water solubility. >350 g/L
6. Chemical safety	Toxic or corrosive, high safety risks.	Moderately safe, irritating at high concentrations.	Generally safe, minimal risks.

**Table 2 foods-13-04048-t002:** Factors and levels analyzed in the 2^3^ factorial design.

Factor	Lower Level (−1)	Upper Level (+1)
Acid concentration (M)	0.5	1
Pulp density (g/L)	5	10
Agitation speed (rpm)	80	120

**Table 3 foods-13-04048-t003:** The 2^3^ experimental design matrix.

Experiment	Acid Concentration (M)	Pulp Density (g/L)	Agitation Speed (rpm)
1	1	1	1
2	1	1	−1
3	1	−1	1
4	1	−1	−1
5	−1	1	1
6	−1	1	−1
7	−1	−1	1
8	−1	−1	−1

**Table 4 foods-13-04048-t004:** Multi-criteria decision-making matrix for organic acid selection. C1: Commercial cost of the acid, C2: potential acid dissociation constant (pKa), C3: compatibility with food matrices, C4: theoretical cadmium leaching capacity in solid matrices, C5: solubility in water, C6: chemical safety. Likert scale evaluations are color-coded: 1 = red, 3 = yellow, 5 = green.

Organic Acid	C1	C2	C3	C4	C5	C6	Reference	Score	Position
Acetic acid	5	3	5	5	5	3	[25,60,61,62,63,64,65,66,67]	4.32	2
Formic acid	3	3	3	3	5	1	[25,60,61,64,65,67,68]	2.78	10
Citric acid	5	3	5	5	5	5	[26,27,28,30,60,61,62,63,64,65,67,68,69,70,71,72,73]	4.71	1
Oxalic acid	5	5	1	5	1	1	[62,64,67,68,72,74]	2.99	9
Fumaric acid	5	3	3	3	1	5	[67,68,75,76]	3.59	6
Succinic acid	1	3	5	3	1	5	[60,64,65,67,68]	3.28	8
Malic acid	3	3	5	3	5	5	[60,62,64,67,68]	4.01	4
Tartaric acid	3	5	5	3	5	5	[26,61,64,65,67,68,72]	4.29	3
Ascorbic acid	1	3	5	3	3	5	[25,26,61,67,68,70,73,77]	3.46	7
Lactic acid	1	3	5	3	5	5	[26,60,64,65,67,68]	3.63	5

**Table 5 foods-13-04048-t005:** Full factorial 2^3^ design matrix for Cd removal.

Experiment	Factor			Cd removal (%)	
	A	B	C	Replicate 1	Replicate 2
5	−1	1	1	54.5	54.2
3	1	−1	1	90.9	90.5
7	−1	−1	1	97	97.1
8	−1	−1	−1	84.8	84.4
6	−1	1	−1	57.6	57.5
2	1	1	−1	36.4	36.8
1	1	1	1	51.5	51.6
4	1	−1	−1	78.8	78.5

**Table 6 foods-13-04048-t006:** Estimated effects for the full factorial 2^3^ design.

Effect	Estimated Effect ± Standard Error
Global mean	68.88 ± 0.82
A: Acid concentration	−9.01 ± 1.64
B: Pulp density	−37.74 ± 1.64
C: Agitation speed	9.06 ± 1.64
AB	−2.86 ± 1.64
AC	4.44 ± 1.64
BC	−3.19 ± 1.64

**Table 7 foods-13-04048-t007:** ANOVA for Cd removal.

Source	Sum of Squares	Degrees of Freedom	Mean Square	F_0_
A	324.901	1	324.901	30.11
B	5696.48	1	5696.48	527.94
C	328.516	1	328.516	30.45
AB	32.7756	1	32.7756	3.04
AC	78.7656	1	78.7656	7.3
BC	40.6406	1	40.6406	3.77
Total Error	86.32	8	10.79	
Total	6588.44	15		

R^2^ = 98.69%; R^2^ (adjusted for d.f.) = 97.54% and F_0_: test statistic.

**Table 8 foods-13-04048-t008:** Optimal values of the factors of the full factorial design 2^3^.

Factor	Optimal Level
A: Acid concentration (M)	(−1) 0.5
B: Pulp density (g/L)	(−1) 5
C: Stirring speed (rpm)	(+1) 120

**Table 9 foods-13-04048-t009:** ORP model parameters.

a	b	c	d	Offset
−23.1083	−0.09615	45.5401	−0.004253	398.7591

**Table 10 foods-13-04048-t010:** Heuristic parameters in the desorption process.

θ (min)	τ (min)	$\lambda =\frac{\theta }{\tau }$
179.89	415.04	2.31

## Data Availability

The original contributions presented in the study are included in the article, further inquiries can be directed to the corresponding author.

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
