# Peer review of "Desorption of Cadmium from Cocoa Waste Using Organic Acids"

_foods, 2024, doi:10.3390/foods13244048_

Round 1
Reviewer 1 Report
Comments and Suggestions for Authors
This study focuses on desorption of Cd from cocoa agro-industrial waste using organic acids. This strategy aims to valorize byproducts for high-value applications, contributing to reduced environmental impact and promoting a circular economy in the cocoa production chain. Citric acid was identified as the most effective organic acid for Cd removal from cacao residues, based on the multi-criteria decision-making (MCDM) matrix results and kinetic desorption studies using different acids and concentrations.
Authors should clarify issues pointed out in the comments given below.
1. Please, describe the novelty more specifically at the end of the Introduction section. Also, a few sentences regarding experimental design should be added.
2. For the sake of clarity, please add flowcharts of cocoa pod husk flour preparation and cadmium desorption experiments in Experimental section.
3. In Table 2, parameters should be uniform, i.e., in both cases should be “Organic acid concentration (M)”, not “Organic acid concentration (M)”in Table 2, and “Acid concentration (M)”in Table 3.
4. Details on instrument type, country and city should be uniformly written.
5. Comparison of obtained results with literature data on cadmium desorption is missing.
6. Explain on the basis of which criteria were chosen the operating conditions for cadmium kinetic experiments (Section 2.5. Cadmium Desorption Kinetics). From literature data or else?
7. Please specify (in caption) that the inset image in Figure 3 refers to 6-24 h period of time.
8. Provide the whole list of all the acids whose criteria were examined, In Section “2.5. Cadmium Desorption Kinetics”, not only those that proved to be the best.
10. Explain the meaning of values 1,3 and 5 in Table 4.
11. For Pareto chart explain in more detail (for example through a graph of the main effects) how it was obtained that certain factors were significant at higher or lower levels.
Author Response
Dear Reviewer 1,
We sincerely thank you for your thorough review and valuable comments on our manuscript titled "Desorption of Cadmium from Cocoa Waste Using Organic Acids." Your insights allowed us identifying areas for improvement and have significantly enhanced the quality of our work.
We have carefully addressed all your comments and suggestions, as outlined below:
Comment 1. Please, describe the novelty more specifically at the end of the Introduction section. Also, a few sentences regarding experimental design should be added.
Answer: Dear Reviewer, we explicitly highlighted the novelty of the study at the end of the Introduction and added details about the experimental design, please check lines 79-81 of the manuscript.
Comment 2. For the sake of clarity, please add flowcharts of cocoa pod husk flour preparation and cadmium desorption experiments in Experimental section.
Answer: We included flowcharts to illustrate the preparation of cocoa pod husk flour and the cadmium desorption experiments in the Experimental section. In the new version of the manuscript these are Figures 1 and 2.
Comment 3. In Table 2, parameters should be uniform, i.e., in both cases should be “Organic acid concentration (M)”, not “Organic acid concentration (M)”in Table 2, and “Acid concentration (M)”in Table 3.
Answer: We ensured that parameter descriptions remain consistent across all tables. Tables 2 and 3, were corrected. The changes were highlighted in red color.
Comment 4. Details on instrument type, country and city should be uniformly written.
Answer: We standardized the format for instrument details throughout the manuscript. We included instrument type, country and city, as suggested by the Reviewer.
Comment 5. Comparison of obtained results with literature data on cadmium desorption is missing.
Answer: Thank you for this suggestion. Upon reviewing the existing literature, we found no studies that specifically address cadmium desorption from solid matrices, such as the cocoa agro-industrial waste examined in our work. Most available research focuses on cadmium desorption from aqueous matrices, which differs significantly from the scope of our study.
While the primary focus of our research is on cadmium desorption in cocoa waste, we identified studies that emphasize alternative methods for cadmium mitigation. These approaches include phytoremediation and treatments applied to soils. For instance, Oñate et al. (2024) explored the use of magnetite-coated nanoparticles (MCNP) for cadmium removal directly from soils.
Additionally, Tariq and Ashraf (2016) demonstrated cadmium removal from soils using hyperaccumulators like sunflowers, achieving 56.03% removal with EDTA application. Similarly, López et al. (2022) reported a 48% reduction in cadmium concentration in cocoa plants by using biochar. These methods, while valuable for long-term soil treatment, do not align with the desorption-focused approach of our study.
The comparison underscores the uniqueness of our research, which provides a targeted, efficient, and scalable solution for cadmium removal in cocoa waste, addressing contamination at a critical point in the cocoa production chain.
References
- López, J., Arroyave, C., Aristizábal, A., Almeida, B., Builes, S., & Chávez, E. (2022). Reducing cadmium bioaccumulation in Theobroma cacao using biochar: Basis for scaling-up to field. Heliyon.
- Oñate, A., Staël, C., Pozo, C., Vera, D., Torres, E., Bolaños, D., & Cumbal, L. (2024). Nanoparticles for treatment of cadmium-contaminated cocoa-growing soils and beans: Performance on metal immobilization and removal. Heliyon.
- Tariq, S., & Ashraf, A. (2016). Comparative evaluation of phytoremediation potential of metal-contaminated soils by four different plant species. Arab Journal of Chemistry, 806-814.
Comment 6. Explain on the basis of which criteria were chosen the operating conditions for cadmium kinetic experiments (Section 2.5. Cadmium Desorption Kinetics). From literature data or else?
Answer: Dear Reviewer, the desorption conditions were selected from previous studies on the metal leaching from soils and ores, as presented in references [25–27, 31]. We clarified the criteria used to select operating conditions in Section 2.5 by adding a new sentence, as presented in lines 146-147.
Comment 7. Please specify (in caption) that the inset image in Figure 3 refers to 6-24 h period of time.
Answer: We updated the caption for Figure 3 to specify that the inset image corresponds to the 6-24 h period.
Comment 8. Provide the whole list of all the acids whose criteria were examined, In Section “2.5. Cadmium Desorption Kinetics”, not only those that proved to be the best.
Answer: We verified that the acid list was complete, and additionally placed an explanatory note at the end of section 2.5. that reads: “Parallel tests were conducted using acetic, tartaric, malic, and lactic acids, each at a concentration of 1 M for 6 hours under identical experimental conditions. An additional test with 0.5 M citric acid was included to evaluate performance at a lower concentration. The acids used here were preselected from the scoring results of the multiparametric priority matrix.”
Comment 9. Explain the meaning of values 1,3 and 5 in Table 4.
Answer: Dear Reviewer, as explained in the methodology, section 2.4, the numbers (1, 3, 5) corresponds to a three-level Likert scale used to evaluate the criterion C1 to C6 in the multiparametric priority matrix.
Comment 10. For Pareto chart explain in more detail (for example through a graph of the main effects) how it was obtained that certain factors were significant at higher or lower levels.
Answer: In the Figure, the blue bars correspond to the factors, where the negative sign corresponds to the lower level from which it was evaluated. Similarly, the orange bar indicates that the factors are statistically significant when evaluated at the higher level. In lines 460 to 462 we added a new paragraph to clarify this fact, as follows:
“In the Figure 7, the blue bars represent the factors, where the negative sign corre-sponds to their lower evaluated levels. Similarly, the orange bar indicates that the fac-tors are statistically significant when evaluated at their higher levels.”
Your feedback has been invaluable in refining the manuscript, and we are confident that these revisions will make the study more robust and clearer to readers.
Thank you once again for your time, effort, and thoughtful suggestions.
Best regards,
The Authors

Reviewer 2 Report
Comments and Suggestions for Authors
Dear authors please see the file in the attachment.

Author Response
Dear Reviewer 2,
We sincerely thank you for your thorough review and valuable comments on our manuscript titled "Desorption of Cadmium from Cocoa Waste Using Organic Acids." Your insights allowed us identifying areas for improvement and have significantly enhanced the quality of our work.
We have carefully addressed all your comments and suggestions, as outlined below:
Comment: “The manuscript is very interesting and very easy to read. It presents a novel approach to desorb Cd from cocoa residues using 6 organic acids. It is a very interesting way to valorise the cocoa residues.
Only a suggestion - It seems to me that there is a lack of a direct correlation between the amount of Cd present in the different types of cocoa analyzed and the amount of Cd present in the specific place where it was collected and perhaps even a relationship between the desorption of Cd, or the amount of Cd as a function of the age of the plants.
The results explanation based on statistical and ORP analysis and are very clear and consistent.”
Answer: Thank you for your kind words about the manuscript and for acknowledging the novelty and clarity of our approach. We appreciate your suggestion regarding the potential correlation between cadmium levels in cocoa and factors such as the specific collection site or the age of the plants.
However, this study is not designed as a georeferenced or agronomic analysis to estimate direct relationships with plant characteristics. Instead, we collected samples from different locations to evaluate cadmium concentrations in cocoa fruits and their distribution within the fruit. This approach aligns with the primary objective of the study, which focuses on the desorption of cadmium from cocoa residues rather than exploring its geospatial or biological correlations. Thank you for your thoughtful feedback.
Comment: “Line 38 – it is more than 4 times not more than twiece.”
Answer: The term was corrected as suggested by the Reviewer.
Comment: “Line 80 – Please check the verb.”
Answer: The verb was checked, and corrected for grammatical purposes.
Comment: “Line 184 – Are the cocoa plants of the same specie? Line 188 – Did the authors compare the Cd desorbed from Cocoa, with the amount of Cd present on soils on the restritct regions studied?”
Answer: Thank you for your thoughtful questions and observations. Regarding the species of cocoa plants (Line 184), the primary objective of this study was to evaluate cadmium in cocoa pods irrespective of the plant species. Since this is neither a georeferenced study nor one focused on specific cocoa plant species, we aimed to generalize our findings across different samples to provide a broader perspective on cadmium desorption.
As for comparing the cadmium desorbed from cocoa with cadmium levels in soils from the restricted regions studied (Line 188), this was beyond the scope of the study. Our focus was on the desorption of cadmium from cocoa residues rather than correlating it with soil cadmium levels or other agronomic factors, such as plant age.
Comment: “Maybe the age of plants could have also an important role on Cd adsorption.
Line 479 – But this is not that we saw in figure 4.”
Answer: Dear Reviewer, Figure 4 illustrates the kinetics of cadmium desorption over time, serving as a preliminary step for designing experiments to determine both the treatment duration and the optimal levels of organic acid concentration. It is important to note that these experiments were not conducted using optimized conditions, as identifying those conditions represents the next step in our methodology.
Dear Reviewer 2, thank you once again for your time, effort, and thoughtful suggestions. We appreciate also your kind words about the manuscript and for acknowledging the novelty and clarity of our approach
Best regards,
The Authors

Round 2
Reviewer 1 Report
Comments and Suggestions for Authors
Dear Colleagues,
I have assessed the revised manuscript ID: foods-3353578: “Desorption of Cadmium from Cocoa Waste using Organic Acids”, authors Sebastián Piedrahíta-Pérez, Juliana Rodríguez-Estrada, Margarita Ramírez-Carmona, Leidy Rendón-Castrillón, Carlos Ocampo-López.
I am satisfied with the corrections made by authors and I find it suitable for publication in Foods.